# Modified Cortisol Circadian Rhythm: The Hidden Toll of Night-Shift Work

**DOI:** 10.3390/ijms26052090

**Published:** 2025-02-27

**Authors:** Aikaterini Andreadi, Stella Andreadi, Federica Todaro, Lorenzo Ippoliti, Alfonso Bellia, Andrea Magrini, George P. Chrousos, Davide Lauro

**Affiliations:** 1Section of Endocrinology and Metabolic Diseases, Department of Systems Medicine, University of Rome Tor Vergata, 00133 Rome, Italy; 2Endocrinology and Diabetology Clinic, Department of Medical Sciences, Foundation Policlinico Tor Vergata, 00133 Rome, Italy; 3Department of Biomedicine and Prevention, University of Rome Tor Vergata, 00133 Rome, Italy; 4Faculty of Medicine, Saint Camillus International University of Health Sciences, 00131 Rome, Italy; 5University Research Institute of Maternal and Child Health and Precision Medicine, Medical School, National and Kapodistrian University of Athens, 11527 Athens, Greece; 6UNESCO Chair on Adolescent Health Care, National and Kapodistrian University of Athens, 11527 Athens, Greece; 7University Research Institute, Choremeion-Aghia Sophia Children’s Hospital, 11527 Athens, Greece

**Keywords:** circadian rhythm, cortisol, metabolic diseases, diabetes

## Abstract

The circadian rhythm of cortisol, a key hormone essential for maintaining metabolic balance and stress homeostasis, is profoundly disrupted by night-shift work. This narrative review examines the physiological mechanisms underlying cortisol regulation, the effects of shift work on its circadian rhythm, the associated health risks, and potential mitigation strategies. Night-shift work alters the natural secretion pattern of cortisol, leading to dysregulation of the hypothalamic–pituitary–adrenal axis, which in turn can contribute to metabolic disorders, cardiovascular diseases, and impaired cognitive function. Understanding the physiological pathways mediating these changes is crucial for developing targeted interventions to mitigate the adverse effects of circadian misalignment. Potential strategies, such as controlled light exposure, strategic napping, and personalized scheduling, may help to stabilize cortisol rhythms and improve health outcomes. This review aims to provide insights that can guide future research and inform occupational health policies for night-shift workers by addressing these challenges.

## 1. Introduction

Circadian rhythms are key regulators of numerous physiological processes, including hormone secretion, metabolism, immune function, and the sleep–wake cycle. These rhythms are governed by an endogenous biological clock located in the suprachiasmatic nucleus (SCN) of the hypothalamus, which synchronizes peripheral clocks in various tissues with environmental cues such as light exposure and food intake [1,2]. Among the hormones regulated by circadian rhythms, cortisol plays a pivotal role in maintaining homeostasis. Secreted by the adrenal glands under the control of the hypothalamic–pituitary–adrenal (HPA) axis, cortisol exhibits a pronounced diurnal pattern. Its levels peak in the early morning, promoting wakefulness, alertness, and metabolic preparedness, before gradually declining throughout the day to facilitate rest and recovery [3,4].

However, lifestyle factors such as night-shift work and irregular sleep patterns can significantly disrupt circadian rhythms. Night-shift work, in particular, misaligns the body’s internal biological clock with external environmental cues, creating a mismatch between physiological processes and the natural day–night cycle [5]. This desynchronization has been linked to adverse metabolic outcomes, including impaired glucose tolerance, dyslipidemia, and an elevated risk of obesity and type 2 diabetes [6,7]. The disruption of the HPA axis and cortisol rhythm further amplifies these effects, underscoring the complex interplay between circadian regulation and metabolic health.

This narrative review examines the physiological mechanisms underlying cortisol regulation and its circadian rhythm, explores how night-shift work disrupts cortisol secretion patterns, and the broader implications for metabolic and overall health. Also, this review assesses the health risks associated with circadian misalignment, particularly focusing on metabolic disorders, cardiovascular disease, and cognitive function.

This manuscript follows this approach, rather than a systematic or scoping review, to provide a comprehensive synthesis of current knowledge on cortisol dysregulation in night-shift workers. A systematic review would require meta-analytical comparisons and predefined inclusion/exclusion criteria, whereas our objective was to integrate diverse findings from endocrinology, occupational health, and circadian biology. Future studies may consider a quantitative meta-analysis to further evaluate specific cortisol-related metrics in night-shift populations [8].

## 2. Cortisol and Circadian Regulation

Cortisol, a key glucocorticoid hormone, exhibits a well-characterized diurnal pattern that is essential for maintaining homeostasis. Synthesized and secreted by the zona fasciculata of the adrenal cortex (Figure 1), cortisol plays a central role in energy metabolism, immune regulation, and the stress response. Its secretion is intricately controlled by the HPA axis, which follows a precise daily rhythm.

Cortisol levels follow a well-defined diurnal rhythm, typically peaking in the early morning and gradually declining throughout the day. This pattern can be assessed through various biospecimens, including saliva, blood, and urine. Salivary cortisol is commonly used due to its non-invasive nature and its ability to reflect biologically active free cortisol levels [9]. Blood cortisol measurements provide total cortisol concentrations, encompassing both free and protein-bound fractions, whereas urinary cortisol offers an integrated measure of cortisol excretion over 24 h [4]. Each method has distinct advantages and limitations, and the choice of assessment depends on research objectives and clinical applications.

To further illustrate these variations, Figure 2 presents a schematic comparison of the diurnal cortisol rhythm in eurhythmic individuals versus those with dysregulated cortisol patterns, such as night-shift workers. The dysregulation is characterized by a blunted or delayed peak and a flattened diurnal decline, often leading to metabolic and cardiovascular disturbances [10].

This rhythmic secretion is regulated by the SCN, often referred to as the brain’s master circadian clock. Located in the hypothalamus, the SCN coordinates physiological processes with external environmental cues, primarily through photic input received from specialized retinal ganglion cells via the retinohypothalamic tract [4]. This mechanism ensures that cortisol secretion remains synchronized with the light–dark cycle. However, disruptions in the SCN or misalignment with external cues can impair this delicate balance, leading to dysregulation of cortisol rhythms [9,11] (Figure 3).

### 2.1. The Cortisol Awakening Response (CAR)

Under normal physiological conditions, cortisol levels exhibit a significant surge within the first 30 to 45 min after awakening, a phenomenon known as the CAR [12] (Figure 4). This peak prepares the body for the upcoming demands of the day by mobilizing energy reserves, enhancing glucose availability, and modulating immune function. Following this morning surge, cortisol levels gradually decline throughout the day, reaching their lowest point in the early nighttime.

Beyond the CAR, additional cortisol-based circadian rhythm metrics provide valuable insights into the temporal dynamics of HPA axis function. Twenty-four-hour cortisol profiling allows for a comprehensive assessment of cortisol secretion patterns throughout the day, capturing variations beyond the morning peak [13]. Acrophase (the timing of peak cortisol secretion) and mesor (the average cortisol level over 24 h) are commonly used in chronobiology research to characterize shifts in circadian timing due to night-shift work [14]. These methods offer a broader perspective on circadian rhythm dysregulation, complementing CAR measurements.

Blunted CAR is characterized by a reduced percentage increase in cortisol levels within 30–45 min of waking. Typically, a healthy CAR results in a 50–150% rise in cortisol levels from the baseline measurement upon waking [9]. A blunted response is defined by an increase of less than 50%, indicating impaired HPA axis function [15] (Figure 5).

### 2.2. The Diurnal Rhythm of Cortisol Is Integral to Several Physiological Functions

Energy Mobilization: cortisol facilitates gluconeogenesis in the liver, ensuring a steady supply of glucose during fasting periods, particularly upon waking.Immune Function: it exerts anti-inflammatory effects, helping to regulate immune system activity and maintain immune homeostasis throughout the day.Stress Adaptation: the circadian rhythm of cortisol optimizes stress responses by promoting an adaptive reaction to acute stressors while preventing prolonged HPA axis activation.

## 3. Impact of Night-Shift Work on Cortisol Rhythms

Night-shift work disrupts the natural circadian rhythm regulated by the SCN, which governs cortisol secretion, essential for energy metabolism, immune modulation, and stress responses. The misalignment caused by night-shift work alters cortisol’s secretion patterns, often resulting in blunted peaks, delayed reactions, and overall dysregulation. Such disruptions are associated with adverse metabolic, cardiovascular, and psychological outcomes, as well as an increased risk of chronic diseases like diabetes and depression. Understanding these changes is essential to mitigate health risks and develop targeted interventions for shift workers [6,10,11,12]. This review focuses on working-age adults (18–65 years), as they represent the primary population affected by night-shift work. Age groups reported in the cited studies have been explicitly stated where applicable to enhance clarity and comparability.

### 3.1. Desynchronization of the SCN

Circadian desynchronization occurs when there is a misalignment between the endogenous circadian system and external environmental cues, often resulting from night-shift work. This misalignment disrupts the normal physiological processes regulated by the circadian clock, leading to adverse health outcomes. Studies have shown that such desynchronization can impair metabolic functions, increase the risk of cardiovascular diseases, and negatively affect cognitive performance [16,17]. Night-shift workers face a profound misalignment between their endogenous circadian rhythms, governed by the SCN, and external work schedules. The SCN in the hypothalamus serves as the body’s master clock, synchronizing physiological processes, including cortisol secretion, to environmental light–dark cycles [11]. Under normal conditions, the SCN regulates a distinct diurnal pattern of cortisol release, characterized by a pronounced peak shortly after waking and a gradual decline throughout the day. However, exposure to artificial light during nighttime and sleep during the day—common in night-shift work—disrupts the SCN’s synchronization with external cues, causing significant desynchronization [10].

Night-shift work is a significant disruptor of circadian synchronization, as it forces individuals to remain active and exposed to artificial light during their biological night, suppressing melatonin secretion and altering the rhythmic release of cortisol [11,18]. This misalignment affects multiple physiological processes, including metabolism, immune function, and cardiovascular regulation, increasing the risk of metabolic syndrome, obesity, and type 2 diabetes [11,19]. Chronically misaligned individuals often exhibit a phase shift in cortisol rhythms, with a delayed or blunted peak in the morning and an inability to properly suppress cortisol levels at night, a pattern commonly observed in night-shift workers [20,21]. Additionally, irregular work patterns—such as rotating shifts or inconsistent night work schedules—exacerbate this disruption, leading to further instability in endocrine and autonomic function [5]. Studies indicate that prolonged exposure to night-shift schedules results in a progressively impaired adaptation, where individuals fail to fully entrain to their altered schedules, contributing to chronic stress and an increased risk of cardiovascular disease [10,22]. These findings highlight the profound impact of night-shift work on the circadian clock and reinforce the need for strategies to mitigate circadian misalignment in occupational settings.

This desynchronization results in a phase shift of cortisol secretion, where peak levels often occur at inappropriate times, such as during the night or early morning, when cortisol is typically low in individuals following a regular day–night cycle [6]. Such disruptions lead to a blunted CAR and a flattened diurnal profile, with cortisol levels remaining relatively elevated across the day and night [9]. The loss of rhythmicity compromises the timing of cortisol secretion and diminishes it is amplitude, reducing the body’s ability to optimize energy metabolism, stress responses, and immune function [23].

Furthermore, circadian rhythm desynchronization is increasingly recognized as a critical factor in disrupting metabolic, immune, and endocrine homeostasis. Exposure to environmental stressors, such as night-shift work and xenobiotics, leads to circadian misalignment, which affects downstream physiological processes and increases disease susceptibility [24]. This misalignment disrupts the synchronization between central and peripheral clocks, influencing key metabolic regulators and hormonal rhythms, such as cortisol secretion patterns. Furthermore, oxidative stress and inflammation play a significant role in the adverse health effects linked to night-shift work, as circadian rhythm misalignment has been associated with increased markers of oxidative damage and impaired recovery processes [25]. These findings highlight the necessity for targeted interventions that mitigate circadian disruption, such as optimized work schedules, light exposure management, and lifestyle modifications to reduce the health risks associated with shift work.

### 3.2. Blunted CAR

The CAR is a hallmark feature of the diurnal rhythm of cortisol secretion; this surge prepares the body for the day’s demands, optimizing energy availability, cognitive function, and stress management. However, studies have shown that night-shift workers frequently exhibit a blunted CAR, which reflects a diminished ability of the HPA axis to adapt to acute stressors effectively [9].

While the CAR is widely used to assess circadian regulation of the HPA axis, additional studies have investigated CAR alterations in night-shift workers. It is demonstrated that shift workers exhibit a delayed and blunted CAR, leading to increased fatigue and cognitive impairment. Similarly, Pruessner et al. [26] found that chronic circadian misalignment results in flatter CAR slopes, suggesting prolonged HPA axis activation. Furthermore, Adam et al. [27] emphasized the role of 24 h cortisol profiling, which provides a broader evaluation of cortisol secretion beyond the morning peak, revealing altered acrophase and mesor in shift workers.

Hypercortisolism is characterized by chronically elevated cortisol levels, often exceeding 15–20 µg/dL in serum or 50–100 µg per 24 h in urine [4]. The clinical diagnosis relies on serum cortisol, salivary late-night cortisol, and 24 h urinary free cortisol (UFC) levels, which help distinguish physiological from pathological hypercortisolism. A persistent dysregulation of cortisol secretion, as seen in night-shift workers, can contribute to metabolic syndrome, hypertension, and cardiovascular disease [10].

Cortisol can be measured using saliva, blood, or urine, each offering distinct advantages. Salivary cortisol is commonly used due to its non-invasive collection and reflection of biologically active free cortisol [9]. Blood cortisol provides total cortisol concentrations, but its levels can fluctuate due to stress-induced acute HPA activation [28]. Urinary free cortisol (UFC) offers an integrated measure of cortisol excretion over 24 h, making it valuable for long-term assessments of HPA axis function [29]. These methodologies contribute to a comprehensive understanding of circadian rhythm integrity and stress-related dysregulation in night-shift workers.

Blunted CAR in night-shift workers may arise from chronic misalignment between the internal circadian clock and external sleep–wake cycles. This misalignment occurs when exposure to artificial light at night interferes with the light-sensitive SCN, delaying the release of melatonin and altering the normal cortisol secretion pattern. Additionally, irregular sleep patterns and shortened daytime sleep reduce the body’s ability to reset its circadian rhythm, further contributing to a disrupted cortisol profile. This desynchronization disrupts the timing of cortisol secretion, often leading to reduced morning peaks or delayed cortisol surges when workers attempt to align their waking hours with irregular schedules [9]. This blunting has been associated with markers of chronic stress, including elevated levels of perceived stress, burnout, and emotional exhaustion [30]. For example, a study by Kudielka et al. [31] found that night-shift workers displayed significantly lower cortisol levels upon waking compared to day-shift workers, along with higher self-reported stress and fatigue levels. This diminished cortisol response was linked to poorer coping mechanisms and increased susceptibility to mood disorders. The diminished CAR may impair cognitive function, mood regulation, and stress resilience, exacerbating night-shift workers’ psychological burden. Existing research indicates that the blue light emitted by these devices can disrupt circadian rhythms and affect cortisol levels. A study found that evening exposure to short-wavelength (blue) light from smartphones without a blue light filter elevated morning cortisol levels and reduced the cortisol awakening response. This suggests that blue light exposure before bedtime may have lingering effects on stress hormone regulation [32].

In addition to its psychological impacts, a blunted CAR has been linked to broader health issues, including impaired glucose metabolism, increased inflammation, and a higher risk of developing cardiovascular diseases, type 2 diabetes, and compromised immune responses. Studies suggest that a blunted CAR correlates with poor immune function, increased inflammation, and greater susceptibility to infections [30]. It also impairs glucose metabolism and lipid regulation, potentially contributing to the heightened risk of type 2 diabetes and cardiovascular diseases observed in shift workers [6]. These findings underscore the importance of understanding and mitigating the impacts of a blunted CAR on both immediate and long-term health outcomes. Potential interventions include promoting consistent sleep schedules, reducing exposure to artificial light at night, and implementing timed light therapy to help realign the circadian clock. Additionally, stress management programs, dietary adjustments, and pharmacological approaches targeting the HPA axis could further support shift workers in mitigating the adverse effects of a blunted CAR.

### 3.3. Chronic Hypercortisolism and Health Risks

Continuous misalignment between the circadian system and external schedules in night-shift workers often leads to chronic hypercortisolism, where cortisol levels persist. Unlike the physiological diurnal rhythm of cortisol secretion, which optimizes metabolic, immune, and stress-related processes, chronic hypercortisolism imposes significant health risks by disrupting these systems. As a biomarker, cortisol’s diurnal rhythm and the CAR reflect the alignment of the HPA axis with the body’s internal clock. Altered cortisol patterns, such as a blunted CAR or a flattened diurnal rhythm, indicate circadian misalignment due to night-shift work [9,14].

Persistent elevation of cortisol levels can lead to metabolic dysregulation, such as increased gluconeogenesis, insulin resistance, and visceral fat deposition [6]. These metabolic changes contribute to the development of conditions like obesity, type 2 diabetes, and metabolic syndrome [4]. Chronic hypercortisolism also promotes dyslipidemia by increasing triglyceride levels and reducing high-density lipoprotein (HDL) cholesterol, further elevating the risk of cardiovascular diseases [33]. Thus, cortisol is not merely a passive marker but a key player in mediating stress-induced endocrine and cardiometabolic alterations.

In addition to metabolic effects, hypercortisolism negatively impacts the cardiovascular system. Elevated cortisol levels increase blood pressure through enhanced sodium retention and vascular reactivity to catecholamines. Over time, this contributes to hypertension, endothelial dysfunction, and atherosclerosis, increasing the risk of stroke and myocardial infarction [23].

The immune system is also impaired by prolonged hypercortisolism. Cortisol’s immunosuppressive effects, while protective against excessive inflammation during acute stress, become detrimental when cortisol levels are chronically elevated. Night-shift workers with chronic hypercortisolism often show reduced immune surveillance, making them more susceptible to infections and slowing recovery from illness [34]. Moreover, chronic low-grade inflammation driven by disrupted cortisol rhythms has been implicated in autoimmune diseases and cancer development [35].

Psychological health is not spared from the effects of chronic hypercortisolism. Elevated cortisol levels over time are strongly associated with mood disorders, including depression and anxiety [36]. Night-shift workers frequently report symptoms of fatigue, irritability, and reduced cognitive performance, which are exacerbated by cortisol dysregulation and disrupted sleep.

This review exclusively focuses on human studies investigating cortisol dysregulation in night-shift workers. Studies were included if they examined cortisol as a biomarker of circadian misalignment or its impact on metabolic, cardiovascular, or psychological health outcomes. We excluded animal studies, in vitro experiments, and studies where cortisol was not assessed as a primary circadian marker. Our aim was to synthesize clinically relevant findings applicable to occupational health and endocrinology [8,37].

## 4. Health Implications of Altered Cortisol Rhythms

### 4.1. Sleep Disorders

Elevated nighttime cortisol levels are a hallmark of various sleep disorders, including insomnia and obstructive sleep apnea (OSA). This hormonal disruption impairs both the initiation and maintenance of sleep, exacerbating sleep fragmentation and reducing overall sleep quality. Research indicates that elevated cortisol levels suppress melatonin secretion—a hormone essential for sleep regulation—thereby delaying sleep onset and increasing the frequency of nighttime awakenings [38].

Chronic disruptions in cortisol rhythms due to sleep disorders are strongly associated with cognitive impairments, including difficulties with attention, memory, and decision-making. Prolonged sleep deprivation amplifies these effects by activating the hypothalamic–pituitary–adrenal (HPA) axis, creating a vicious cycle of stress and poor sleep. This feedback loop results in heightened fatigue, diminished work performance, and reduced quality of life [39].

Furthermore, altered cortisol rhythms contribute significantly to metabolic dysregulation. Elevated nighttime cortisol levels promote glucose production and reduce insulin sensitivity, thereby increasing the risk of developing type 2 diabetes and obesity. Sleep disorders exacerbate these metabolic risks by promoting chronic low-grade inflammation, driven by persistent HPA axis activation [40].

The cardiovascular consequences of altered cortisol rhythms are equally profound. Elevated nighttime cortisol levels are associated with increased blood pressure, heart rate variability, and a heightened risk of developing hypertension and cardiovascular disease. Individuals with sleep disorders experience these effects more acutely due to nocturnal overactivation of the sympathetic nervous system, a condition worsened by disrupted cortisol rhythms [11,41].

Cortisol dysregulation also impacts mental health. Elevated cortisol levels are strongly linked to mood disorders such as depression and anxiety. The bidirectional relationship between sleep and mental health implies that poor sleep exacerbates mood disorders, while psychological stress further disrupts sleep patterns and cortisol regulation [42].

Addressing these disruptions is crucial for improving patient outcomes. Treatment strategies for sleep disorders often focus on restoring normal cortisol rhythms. Cognitive-behavioral therapy for insomnia (CBT-I) has proven effective in lowering nighttime cortisol levels by addressing maladaptive sleep behaviors and thought patterns. Additionally, pharmacological interventions, such as melatonin supplementation, counteract the suppressive effects of cortisol on natural sleep-promoting mechanisms [43]. Given the widespread health implications of altered cortisol rhythms, targeted therapeutic approaches are essential for mitigating their effects on cognitive function, metabolic health, cardiovascular regulation, and mental well-being.

### 4.2. Metabolic Dysregulation

Cortisol plays a pivotal role in glucose metabolism by stimulating gluconeogenesis in the liver, ensuring adequate glucose availability during stress. However, chronic hypercortisolism or flattened cortisol rhythms disrupt this balance, leading to hyperglycemia and insulin resistance. Insulin resistance occurs when target tissues, such as muscle and adipose tissue, fail to respond effectively to insulin, reducing glucose uptake and utilization. Persistent hyperglycemia places additional stress on pancreatic beta cells, impairing insulin secretion and worsening glucose intolerance [44].

Dysregulation of the HPA axis is a hallmark of metabolic syndrome. Individuals with flattened cortisol rhythms or elevated evening cortisol levels demonstrate reduced insulin sensitivity and an increased risk of developing T2DM [4]. This pattern is particularly evident in individuals with obesity, where excess adipose tissue acts as an endocrine organ, releasing pro-inflammatory cytokines that further impair insulin signaling. Elevated cortisol levels also promote lipolysis, increasing circulating free fatty acids, which further disrupt glucose transport and exacerbate insulin resistance [45].

Obesity is closely linked to altered cortisol metabolism. In visceral adipose tissue, the enzyme 11β-hydroxysteroid dehydrogenase type 1 (11β-HSD1) converts inactive cortisone to active cortisol, thereby amplifying local cortisol concentrations. This localized increase contributes to adipocyte hypertrophy, impaired adipokine secretion, and systemic metabolic dysfunction [44]. Obesity-induced inflammation further dysregulates the HPA axis, creating a feedback loop of metabolic impairment.

Chronic stress and disrupted sleep patterns, both prevalent in modern lifestyles, significantly alter cortisol rhythms and increase the risk of T2DM. Night-shift workers and individuals with sleep disorders often exhibit blunted morning cortisol responses and elevated evening cortisol levels, impairing nocturnal glucose regulation and predisposing them to fasting hyperglycemia and glucose intolerance [11].

Effective therapeutic strategies to address cortisol-related metabolic dysregulation include lifestyle modifications such as stress management, improved sleep hygiene, and regular physical activity. Emerging pharmacological interventions targeting the HPA axis, such as glucocorticoid receptor antagonists and 11β-HSD1 inhibitors, also show promise in mitigating cortisol-driven metabolic abnormalities [46].

Cortisol plays a dual role in metabolic regulation, with physiological rhythms essential for maintaining glucose and insulin homeostasis. However, chronic disruption of these rhythms contributes to significant metabolic derangements, including insulin resistance, obesity, and diabetes mellitus type 2 (T2DM). Addressing the underlying causes of HPA axis dysregulation holds the potential for mitigating the metabolic consequences of altered cortisol activity.

### 4.3. Cardiovascular Diseases

Cortisol directly impacts blood pressure regulation by affecting vascular tone and sodium retention. It enhances the sensitivity of vascular smooth muscle cells to catecholamines, promoting vasoconstriction and increasing systemic vascular resistance [47]. Furthermore, cortisol stimulates the activity of the mineralocorticoid receptor, mimicking aldosterone-like effects that lead to increased sodium reabsorption and water retention in the kidneys. This raises the intravascular volume and blood pressure over time [48].

Studies on night-shift workers reveal a strong correlation between disrupted cortisol rhythms and the prevalence of hypertension. Altered sleep–wake cycles result in a blunted cortisol awakening response and elevated nocturnal cortisol levels, associated with increased nighttime blood pressure and diminished nocturnal dipping—both predictors of cardiovascular risk [11]. Prolonged exposure to elevated cortisol also impairs endothelial function, reducing nitric oxide availability and promoting vascular stiffness, further exacerbating hypertension [47].

Cortisol contributes to the pathogenesis of atherosclerosis through multiple pathways. It promotes low-grade chronic inflammation, a key driver of plaque formation, by upregulating pro-inflammatory cytokines such as interleukin-6 (IL-6) and tumor necrosis factor-alpha (TNF-α) [49]. Elevated cortisol also enhances the oxidation of low-density lipoprotein (LDL) cholesterol, a crucial step in forming atherosclerotic plaques [49].

In addition to its direct effects on the vasculature, cortisol impairs lipid metabolism, increasing levels of triglycerides and LDL cholesterol while reducing high-density lipoprotein (HDL) cholesterol [50]. These dyslipidemic effects further accelerate the development of atherosclerosis. Night-shift workers, who often experience chronic stress and disrupted sleep, exhibit higher cortisol levels and an increased risk of atherosclerotic events, including myocardial infarction and stroke [50].

Cortisol’s role in cardiac remodeling and arrhythmogenesis has also been studied. Chronic cortisol exposure induces hypertrophy of cardiac myocytes and fibrosis of the myocardium, leading to impaired cardiac function and increased susceptibility to heart failure [23]. Additionally, elevated cortisol levels are linked to autonomic dysfunction, characterized by heightened sympathetic activity and reduced parasympathetic tone, which contribute to arrhythmias and other cardiovascular complications [47].

Addressing cortisol dysregulation is essential for reducing the cardiovascular risks associated with chronic stress and disrupted circadian rhythms. Interventions such as stress management, improving sleep hygiene, and regular physical activity have lowered cortisol levels and improved cardiovascular outcomes [51]. Pharmacological approaches, including glucocorticoid receptor antagonists and 11β-HSD1 inhibitors, are being explored as potential therapies for mitigating cortisol-induced cardiovascular damage [46].

Elevated cortisol levels, particularly in night-shift workers, play a significant role in the development of hypertension, atherosclerosis, and other cardiovascular diseases. The mechanisms involve a combination of direct vascular effects, lipid metabolism impairment, and promotion of systemic inflammation. Effective management of cortisol dysregulation offers a promising avenue for reducing cardiovascular morbidity and mortality.

### 4.4. Mental Health Disorders

Chronic stress and elevated cortisol levels are strongly linked to anxiety disorders. Cortisol increases the reactivity of the amygdala, a brain region responsible for fear and stress responses, leading to hypervigilance and exaggerated stress reactions characteristic of anxiety [52,53].

Shift workers, who often experience circadian misalignment, are at higher risk for anxiety. Studies indicate that altered cortisol rhythms, such as a flattened diurnal slope or elevated evening cortisol levels, correlate with increased anxiety symptoms [20]. This hormonal dysregulation intensifies hyperactivity in the HPA axis, creating a self-perpetuating cycle of heightened stress and anxiety [35].

Cortisol also plays a complex role in depression, with both hypercortisolism and hypocortisolism observed in affected individuals. Hypercortisolism, commonly linked to melancholic depression, is characterized by persistently elevated basal cortisol levels and impaired HPA axis feedback regulation [54]. Chronic cortisol elevation leads to hippocampal damage, reducing neurogenesis and impairing memory and emotional regulation [6]. Conversely, hypocortisolism, typically associated with atypical depression, results in an insufficient stress response and diminished resilience to psychosocial stressors [55].

Shift workers often present with hypercortisolism due to chronic stress and sleep disturbances, which increases their risk of depressive symptoms. Studies have identified elevated evening cortisol levels and blunted morning rises as biomarkers of major depressive disorder [56]. These disrupted cortisol patterns further exacerbate depressive states by impairing the brain’s capacity to regulate mood and stress responses effectively.

Mood disorders, including bipolar disorder, are also strongly linked to cortisol dysregulation. In bipolar disorder, manic episodes are often accompanied by heightened cortisol levels, whereas depressive episodes are associated with diminished HPA axis activity [52]. Circadian misalignment, a common issue for shift workers, exacerbates these fluctuations, destabilizing mood regulation mechanisms [18].

Cortisol affects mental health through its interactions with the brain’s neurocircuitry. Prolonged cortisol exposure reduces the prefrontal cortex’s and hippocampus’s structural and functional integrity, impairing decision-making, emotional regulation, and memory [55,57]. Additionally, cortisol’s effects on the monoaminergic system, including serotonin and dopamine pathways, contribute to dysregulated mood and emotional states [58].

Addressing cortisol dysregulation is essential for improving mental health outcomes in shift workers. Lifestyle interventions, such as stress management techniques, mindfulness practices, and regular physical activity, have been shown to help regulate cortisol secretion and alleviate symptoms of anxiety and depression [23]. Pharmacological approaches, including glucocorticoid receptor antagonists and circadian rhythm stabilizers like melatonin, are also promising strategies for mitigating cortisol-related mental health issues [51].

Also, it is shown that shift work and circadian misalignment can adversely affect sexual health. For instance, a study reported that men with Shift Work Sleep Disorder (SWSD) had significantly lower erectile function scores compared to those without SWSD. Men working night shifts exhibited even poorer erectile function. The study suggests that circadian rhythm disturbances may significantly impact erectile function [59].

Similarly, research focusing on midwives indicated that those working night shifts were more likely to experience reproductive problems and sexual dysfunctions. The most pronounced differences were observed in infertility rates and the number of miscarriages. The study concluded that shift work negatively affects reproductive and sexual health, emphasizing the need for work schedules that allow for adequate rest and social life [60].

Altered cortisol secretion significantly contributes to anxiety, depression, and mood disorders, particularly in shift workers who are prone to circadian misalignment. The bidirectional relationship between cortisol dysregulation and mental health underscores the importance of addressing HPA axis dysfunction through targeted interventions aimed at enhancing emotional well-being and reducing the impact of chronic stress.

This review specifically focuses on the impact of night-shift work-induced cortisol dysregulation on metabolic, cardiovascular, and psychological health outcomes. While there is growing evidence linking circadian misalignment to increased cancer risk, this review does not explore oncological aspects, as our objective is to examine the physiological consequences of altered cortisol rhythms in night-shift workers. Future studies may further investigate the role of circadian disruption in tumorigenesis, integrating cortisol as a potential mediator in cancer-related mechanisms [61].

## 5. Future Research Directions

Future research should prioritize advancing personalized interventions that account for individual variations in circadian adaptability and cortisol responses. Circadian rhythms and cortisol secretion patterns vary significantly across individuals, and identifying reliable biomarkers for these differences could greatly enhance the precision of therapeutic strategies. Genetic predispositions, lifestyle factors, and environmental influences are likely key determinants of intervention effectiveness and warrant deeper investigation to optimize tailored treatments.

Longitudinal studies are critical to evaluating the long-term efficacy and sustainability of various interventions across diverse populations. Future research should account for demographic variations such as age, gender, socioeconomic status, and cultural background to ensure that evidence-based guidelines are both comprehensive and adaptable. This inclusive approach would allow for the development of universally applicable recommendations while accommodating the unique needs of specific groups.

Integrating advanced technologies, such as wearable devices and artificial intelligence, could transform the monitoring and management of circadian and cortisol-related disorders. These tools have the potential to provide real-time feedback on physiological changes, thereby enhancing the customization of interventions and improving patient adherence. Exploring the synergy between pharmacological and non-pharmacological treatments—including behavioral therapies, dietary modifications, exercise, and light exposure—could further optimize therapeutic outcomes.

Moreover, the impact of circadian and cortisol dynamics on chronic diseases, such as metabolic syndrome, cardiovascular disorders, and mental health conditions, remains a relatively underexplored area. Investigating these relationships could uncover novel insights into disease prevention and management. This underscores the importance of interdisciplinary collaboration, drawing from fields such as endocrinology, neurology, psychology, and public health, to address the multifaceted nature of circadian disruptions and cortisol dysregulation.

By prioritizing these research directions, the field can progress toward holistic, patient-centered solutions that better address the complexities of individual health trajectories. A more nuanced understanding of circadian and cortisol biology will pave the way for innovative approaches to improving both short- and long-term health outcomes across a wide range of populations.

## 6. Conclusions

Night-shift work significantly disrupts the natural circadian rhythm of cortisol secretion, resulting in a range of physical and mental health consequences. This misalignment of biological rhythms heightens the risk of metabolic disorders, cardiovascular diseases, and cognitive impairments, while also increasing susceptibility to mood disorders such as depression and anxiety.

Effectively addressing these challenges requires a comprehensive, multi-faceted approach that integrates evidence-based interventions with workplace modifications. Practical strategies, including tailored light exposure, optimized sleep schedules, and stress management techniques, can help mitigate circadian misalignment and reduce the adverse effects associated with night-shift work. Incorporating regular breaks, promoting healthier lifestyle choices, and encouraging physical activity can further enhance these interventions.

Raising awareness among employers and workers about the health risks of circadian disruption is equally critical. Implementing workplace policies that prioritize the well-being of night-shift employees—such as providing access to wellness programs, improving shift rotations, and offering flexible scheduling—can significantly improve both quality of life and productivity.

Additionally, ongoing research and innovation are essential for refining these interventions to maximize their efficacy across different industries and populations. Developing personalized solutions based on individual circadian profiles, leveraging wearable technologies for real-time monitoring, and exploring the use of pharmacological aids could further enhance outcomes. By adopting a proactive and holistic approach, organizations can foster healthier work environments, ensuring long-term health benefits for night-shift workers.

## Figures and Tables

**Figure 1 ijms-26-02090-f001:**
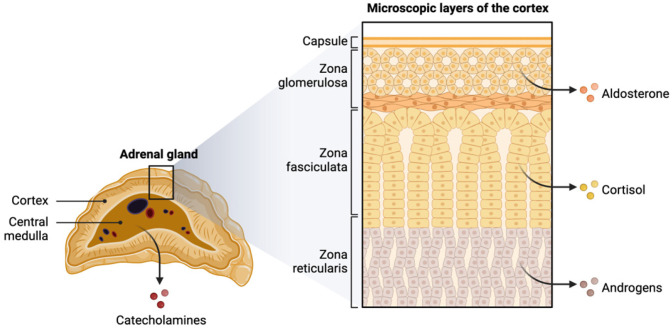
Structure of the adrenal gland, highlighting the cortex and medulla. The cortex is divided into three microscopic layers: the zona glomerulosa, which produces aldosterone; the zona fasciculata, which synthesizes cortisol; and the zona reticularis, responsible for androgen production. The central medulla generates catecholamines such as adrenaline and noradrenaline.

**Figure 2 ijms-26-02090-f002:**
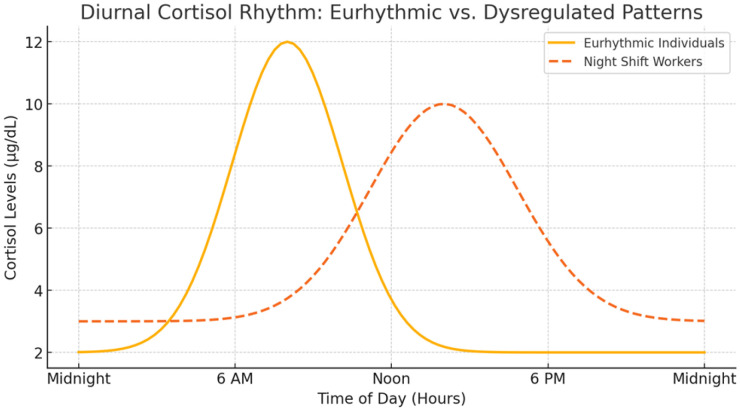
Comparison of diurnal cortisol rhythms between eurhythmic individuals and night-shift workers: Across different biospecimen types (urine, saliva, plasma), the general pattern remains similar; however, amplitude and timing may vary. Dysregulation in night-shift workers leads to a phase shift and altered acrophase (peak timing) and mesor (average cortisol levels). At least four time points across the day (e.g., morning, midday, evening, and night) are typically required to reliably capture this rhythm.

**Figure 3 ijms-26-02090-f003:**
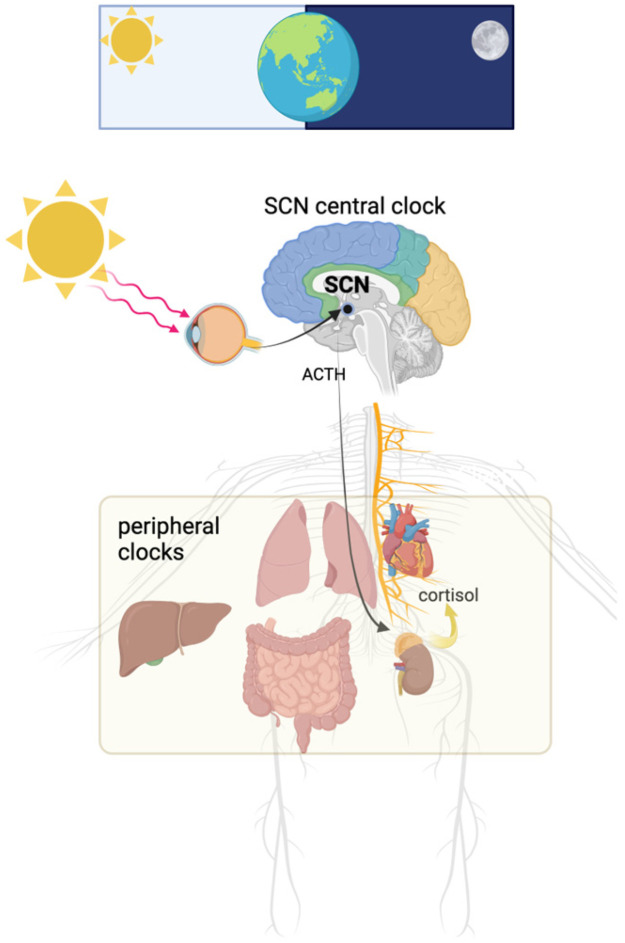
The wake/sleep cycle and organismal feeding/fasting are linked to the solar cycle. Through light stimulation of the retina, a central CNS circadian clock in the suprachiasmatic nucleus is related to organisms’ active/rest cycles. The central clock synchronizes Individual cellular peripheral clocks in almost every tissue, increasing circulating cortisol by stimulating the pituitary production of adrenocorticotropic hormone (ACTH).

**Figure 4 ijms-26-02090-f004:**
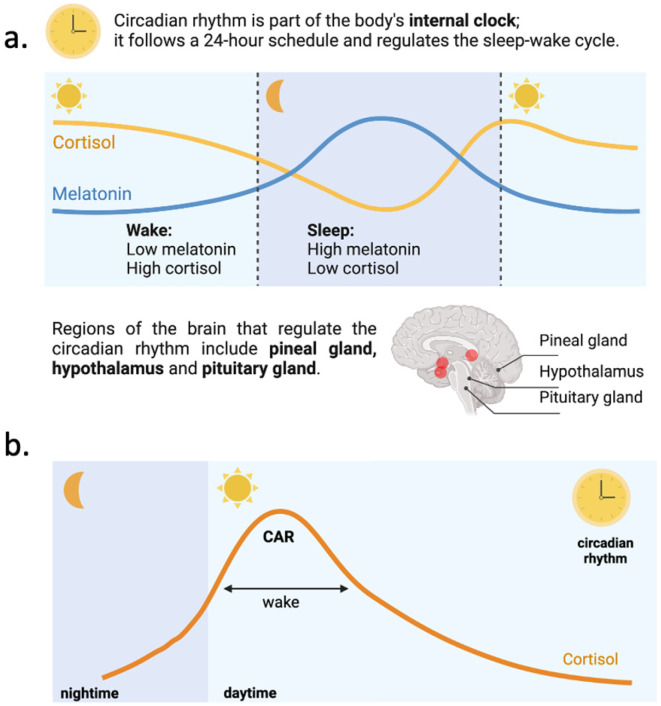
(**a**) Circadian rhythm, which is part of the body’s internal clock regulating the sleep–wake cycle. It shows the inverse relationship between cortisol (higher during the day) and melatonin (higher at night). During wakefulness, melatonin is low, and cortisol is high, while the opposite occurs during sleep. Brain regions involved in circadian rhythm regulation include the pineal gland, hypothalamus, and pituitary gland; (**b**) the figure depicts the daily circadian rhythm of cortisol levels, highlighting the CAR. Cortisol rises rapidly after waking, peaks during the early daytime, and gradually declines throughout the day, reaching its lowest levels at nighttime. This pattern reflects the role of cortisol in preparing the body for activity and maintaining energy balance. The rhythm is regulated by the body’s internal clock.

**Figure 5 ijms-26-02090-f005:**
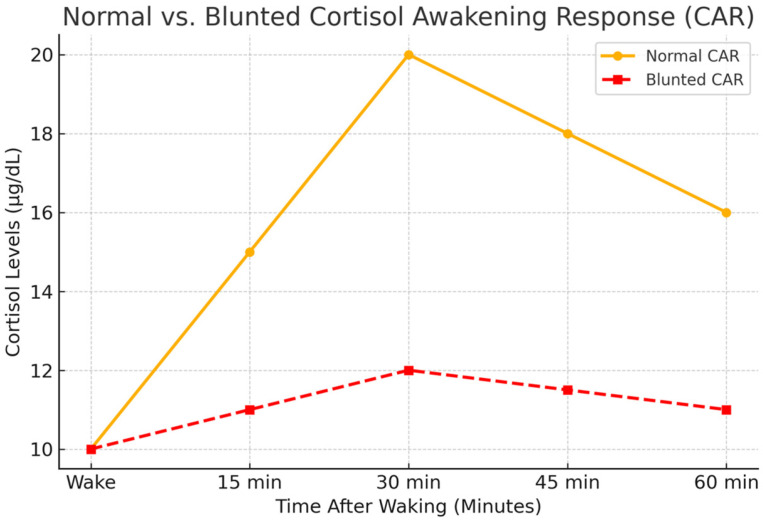
Comparison of normal vs. blunted CAR patterns in shift workers: The CAR follows a predictable increase in cortisol levels within 30–45 min post-awakening. Biospecimen type (saliva, blood) influences measured levels but not the overall trend. A blunted CAR is characterized by reduced peak response and lower total cortisol output, often linked to chronic stress or fatigue. To accurately assess CAR, at least three samples (e.g., at waking, 30 min, and 60 min) are recommended.

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
