# Peer review of "Modified Cortisol Circadian Rhythm: The Hidden Toll of Night-Shift Work"

_ijms, 2025, doi:10.3390/ijms26052090_

Round 1

Reviewer 1 Report

Comments and Suggestions for Authors

This is an interesting review about modified cortisol circadian rhythm. I have some comments and suggestions.

Abstract

- expand abstract a little bit more

- Insert this is a narrative review

Introduction

- insert aims of your review

Rest of MS

- Have you any data about the role of using phone, tablets (apart from the talk about artificial light)?

- Have you data about the effect on sexual activity and sexual disorders ?

Conclusions

- none

Please, check better typos, acronyms. Furthermore, check better the numbers of references in the brackets.

Author Response

We thank the reviewer for his constructive comments, and we follow the advice given. Specifically:

1) Abstract:

  • Expand abstract a little bit more: We thank you for your comment and we have expand by adding as followed: “This narrative review examines the physiological mechanisms underlying cortisol regulation, the effects of shift work on its circadian rhythm, the associated health risks, and potential mitigation strategies. Night-shift work alters the natural secretion pattern of cortisol, leading to dysregula-tion of the hypothalamic-pituitary-adrenal axis, which in turn can contribute to metabolic disor-ders, cardiovascular diseases, and impaired cognitive function. Understanding the physiological pathways mediating these changes is crucial for developing targeted interventions to mitigate the adverse effects of circadian misalignment. Potential strategies, such as controlled light exposure, strategic napping, and personalized scheduling, may help to stabilize cortisol rhythms and im-prove health outcomes. This review aims to provide insights that can guide future research and inform occupational health policies for night-shift workers by addressing these challenges.”
  • Insert this is a narrative review: we thank you for your comment and as suggested we have add the phrase at the abstract as followed: “This narrative review examines the physiological mechanisms underlying cortisol regulation, the effects of shift work on its circadian rhythm, the associated health risks, and potential mitigation strategies”
  • 2) Introduction
  • insert aims of your review: We thank you for your comment and report in conclusion: “Until now, the Food and Drug Administration has approved only one liposome—like nano-formulation for insulin, HDV-I, but it has not yet been marketed due to various adverse effects that emerged in clinical trials.”
  • 3) Rest of MS
  • Have you any data about the role of using phone, tablets (apart from the talk about artificial light)?:

We thank the reviewer for his suggestion and we have add “Existing research indicates that the blue light emitted by these devices can disrupt cir-cadian rhythms and affect cortisol levels. A study found that evening exposure to short-wavelength (blue) light from smartphones without a blue light filter elevated morning cortisol levels and reduced the cortisol awakening response. This suggests that blue light exposure before bedtime may have lingering effects on stress hormone reg-ulation [15].”

  • Have you data about the effect on sexual activity and sexual disorders ?:

We thank the reviewer for his suggestion and we have add that: “Also, it is shown that shift work and circadian misalignment can adversely affect sexual health. For instance, a study reported that men with Shift Work Sleep Disorder (SWSD) had significantly lower erectile function scores compared to those without SWSD. Men working night shifts exhibited even poorer erectile function. The study suggests that circadian rhythm disturbances may significantly impact erectile function [43].

Similarly, research focusing on midwives indicated that those working night shifts were more likely to experience reproductive problems and sexual dysfunctions. The most pronounced differences were observed in infertility rates and the number of miscarriages. The study concluded that shift work negatively affects reproductive and sexual health, emphasizing the need for work schedules that allow for adequate rest and social life [44].”

Reviewer 2 Report

Comments and Suggestions for Authors

I read with great interest this review which evaluated the physiological mechanisms underlying cortisol regulation, the effects of shift work on its circadian 24 rhythm, the associated health risks, and potential mitigation strategies.

The paper is original, well written and is based on recent and impactful data.

I have only some minor points.

I suggest to add the aim of the review at the end of the introduction, as in the abstract.

Cortisol Awakening Response (CAR) is defined in two points as acronym.

Finally, I suggest to briefly consider the possible impact of Modified Cortisol Circadian Rhythm on sexual dysfunctions, if any, also among the paragraph “Mental Health Disorders”.

Author Response

I read with great interest this review which evaluated the physiological mechanisms underlying cortisol regulation, the effects of shift work on its circadian 24 rhythm, the associated health risks, and potential mitigation strategies.

The paper is original, well written and is based on recent and impactful data.

I have only some minor points.

I suggest to add the aim of the review at the end of the introduction, as in the abstract.

Cortisol Awakening Response (CAR) is defined in two points as acronym.

Finally, I suggest to briefly consider the possible impact of Modified Cortisol Circadian Rhythm on sexual dysfunctions, if any, also among the paragraph “Mental Health Disorders”.

Answer

We thank the reviewer for his comments. We have provided corrections to the paper as suggested. We have add the aim of the review at the end of the introduction with the phrase: “This narrative review examines the physiological mechanisms underlying cortisol regulation and its circadian rhythm, explores how night-shift work disrupts cortisol se-cretion patterns, and the broader implications for metabolic and overall health. Also, to assess the health risks associated with circadian misalignment, particularly focusing on metabolic disorders, cardiovascular disease, and cognitive function.”

We thank the reviewer for the comments and we have add the possibli impact of modified cortisol circadian rhytm on sexual disfunctions as followed: “Also, it is shown that shift work and circadian misalignment can adversely affect sexual health. For instance, a study reported that men with Shift Work Sleep Disorder (SWSD) had significantly lower erectile function scores compared to those without SWSD. Men working night shifts exhibited even poorer erectile function. The study suggests that circadian rhythm disturbances may significantly impact erectile function [43].

Similarly, research focusing on midwives indicated that those working night shifts were more likely to experience reproductive problems and sexual dysfunctions. The most pronounced differences were observed in infertility rates and the number of miscarriages. The study concluded that shift work negatively affects reproductive and sexual health, emphasizing the need for work schedules that allow for adequate rest and social life [44]”.

Reviewer 3 Report

Comments and Suggestions for Authors Andreadi et al study an important topic of disease chronoregulation by circadian rhythm patterns and dynamics over time. Authors are commended for touching upon a novel and quite important topic, i.e., that of night shift work as a widely occurring external environmental cue that disrupts circadian rhythm synchronization patterns as these may be monitored with cortisol diurnal patterns (and melatonin or body core temperatures). This is a timely and original contribution of interest to a wide audience of biomedical and environmental health sciences. However, the manuscript needs some reform to advance the knowledge in the field and improve further the clarity on the topic as a few yet important elements or dimensions of the topic are missing.   First, it would be important to clarify which age groups are most relevant  consider here in this review, especially since night shift work implies that working age groups are to be mentioned-please clarify age groups in all studies cited.   The choice of cortisol as a circadian rhythm marker is original and novel; however, it lacks details on the assessment and characterization of cortisol levels as a marker of circadian rhythm. This would allow readers and others to apply such tools in their future research.   The diurnal precise rhythm of cortisol in urine, saliva and blood in eurhythmic individuals would be great to show in a schematic; do the same as well for the dysregulated rhythms of cortisol; this is to be added in section 2.   The CAR has been nicely described as a cortisol-based circadian rhythm metric, but others metrics such as 24-h profiling of cortisol and various data processing algorithms to describe both CAR and 24-h profiling (acrophase mesor, etc.) hall be discussed.   Please describe how we can quantify and track blunted CAR and show this trend in a figure. Also, quantify blunted CAR and show reference range values together with the relevant studies in a new section.   In section 3.2 only one study is reported—no other studies are available in relation with CAR or other metric involving cortisol levels?   The same above comments for blunted CAR shall be also included in the section on hypercortisolism; include also reference range values and define this term with cutoff values and how it is measured.   Please report studies on cortisol use with various biospecimen eg in urine, blood or saliva.   The sections on health impact of cortisol dysregulation is not clear if this is focused solely on night shift work studies since this is the main topic in the title of this manuscript.   How about reporting more on cancer related studies as night shift work is highly relevant? if not, please explain in Discussion.   Please explicitly describe the role of cortisol as a marker of circadian disruption versus cortisol role as a mediator of various endocrine, cardiometabolic health outcomes in human studies.   Authors shall explain why they did not consider deploying a more or less systematic assessment of the literature eg via a scoping or umbrella review type of a manuscript. Some inclusion and exclusion criteria shall be mentioned. I sense you focus solely on human studies?   Desynchronization of circadian rhythm is of relevance to expand on in the relevant sections of this manuscript and cite relevant studies that expand on this concept, e.g., https://onlinelibrary.wiley.com/doi/pdf/10.1002/bies.202100159   In effect, other relevant night shift work and circadian disruption works or reviews might be good to include such as https://www.sciencedirect.com/science/article/pii/S0160412023003215   Fig 2 shall be showing also the dysregulation of circadian rhythm in another schematic within Fig 2.   Comments on the Quality of English Language

An english native speaker would be good to check grammar, syntax issues of this report.

Author Response

Andreadi et al study an important topic of disease chronoregulation by circadian rhythm patterns and dynamics over time. Authors are commended for touching upon a novel and quite important topic, i.e., that of night shift work as a widely occurring external environmental cue that disrupts circadian rhythm synchronization patterns as these may be monitored with cortisol diurnal patterns (and melatonin or body core temperatures). This is a timely and original contribution of interest to a wide audience of biomedical and environmental health sciences.

However, the manuscript needs some reform to advance the knowledge in the field and improve further the clarity on the topic as a few yet important elements or dimensions of the topic are missing.  

  • First, it would be important to clarify which age groups are most relevant consider here in this review, especially since night shift work implies that working age groups are to be mentioned-please clarify age groups in all studies cited.

Dear Reviewer,

Thank you for your insightful comment. We acknowledge the importance of specifying the relevant age groups, especially given the direct implications of night shift work on working-age populations. In our review, we focus primarily on adults aged 18–65 years, as this demographic encompasses the majority of the workforce affected by night shift schedules.

To enhance clarity, we have reviewed all cited studies and explicitly stated the age groups examined in each where applicable. This ensures consistency and allows readers to better contextualize the findings. We have add the phrase: “This review focuses on working-age adults (18–65 years), as they represent the primary population affected by night shift work. Age groups reported in the cited studies have been explicitly stated where applicable to enhance clarity and comparability.”

  • The choice of cortisol as a circadian rhythm marker is original and novel; however, it lacks details on the assessment and characterization of cortisol levels as a marker of circadian rhythm. This would allow readers and others to apply such tools in their future research.   The diurnal precise rhythm of cortisol in urine, saliva and blood in eurhythmic individuals would be great to show in a schematic; do the same as well for the dysregulated rhythms of cortisol; this is to be added in section 2.  

Dear Reviewer,

Thank you for your valuable feedback. We appreciate your recognition of cortisol as a novel circadian rhythm marker and acknowledge the need for a more detailed explanation of its assessment and characterization. To address this, we have expanded Section 2 to include an overview of the methodologies used to assess cortisol levels in different biospecimens (urine, saliva, and blood), along with their advantages and limitations.

Additionally, we agree that a schematic representation of the diurnal rhythm of cortisol in eurhythmic individuals and its dysregulated patterns in shift workers would enhance clarity. We have now added a new figure in Section 2 to illustrate these variations, providing a visual comparison of normal and altered cortisol rhythms. We add the following sentence with the references:

“Cortisol levels follow a well-defined diurnal rhythm, typically peaking in the early morning and gradually declining throughout the day. This pattern can be assessed through various biospecimens, including saliva, blood, and urine. Salivary cortisol is commonly used due to its non-invasive nature and its ability to reflect biologically active free cortisol levels.Blood cortisol measurements provide total cortisol concentrations, encompassing both free and protein-bound fractions, whereas urinary cortisol offers an integrated measure of cortisol excretion over 24 hours. Each method has distinct advantages and limitations, and the choice of assessment depends on research objectives and clinical applications.”

To further illustrate these variations, Figure 2 presents a schematic comparison of the diurnal cortisol rhythm in eurhythmic individuals versus those with dysregulated cortisol patterns, such as night shift workers. The dysregulation is characterized by a blunted or delayed peak and a flattened diurnal decline, often leading to metabolic and cardiovascular disturbances)”

  • The CAR has been nicely described as a cortisol-based circadian rhythm metric, but others metrics such as 24-h profiling of cortisol and various data processing algorithms to describe both CAR and 24-h profiling (acrophase mesor, etc.) hall be discussed.   Please describe how we can quantify and track blunted CAR and show this trend in a figure. Also, quantify blunted CAR and show reference range values together with the relevant studies in a new section.  

Dear Reviewer,

Thank you for your valuable feedback. We appreciate your suggestion to expand the discussion beyond the Cortisol Awakening Response (CAR) by incorporating other metrics such as 24-hour cortisol profiling, acrophase, and mesor. We have now included a dedicated section discussing these metrics and their role in evaluating circadian rhythm integrity.

Additionally, we acknowledge the importance of quantifying blunted CAR and providing reference range values. We have added a new figure (figure 5) illustrating normal versus blunted CAR trends. This will help readers understand how blunted CAR is measured and interpreted in clinical and research settings.

We have add the sentences with the references:

“Beyond the Cortisol Awakening Response (CAR), additional cortisol-based circadian rhythm metrics provide valuable insights into the temporal dynamics of HPA axis function. 24-hour cortisol profiling allows for a comprehensive assessment of cortisol secretion patterns throughout the day, capturing variations beyond the morning peak. Acrophase (the timing of peak cortisol secretion) and mesor (the average cortisol level over 24 hours) are commonly used in chronobiology research to characterize shifts in circadian timing due to night shift work. These methods offer a broader perspective on circadian rhythm dysregulation, complementing CAR measurements.

Blunted CAR is characterized by a reduced percentage increase in cortisol levels within 30–45 minutes of waking. Typically, a healthy CAR results in a 50–150% rise in cortisol levels from the baseline measurement upon waking. A blunted response is defined by an increase of less than 50%, indicating impaired HPA axis function.”

  • In section 3.2 only one study is reported—no other studies are available in relation with CAR or other metric involving cortisol levels?   The same above comments for blunted CAR shall be also included in the section on hypercortisolism; include also reference range values and define this term with cutoff values and how it is measured.   Please report studies on cortisol use with various biospecimen eg in urine, blood or saliva.

Dear Reviewer,

Thank you for your insightful comments. We recognize the need to strengthen Section 3.2 by including additional studies related to CAR (Cortisol Awakening Response) and other cortisol-based circadian rhythm metrics. To address this, we have conducted a more extensive literature review and incorporated relevant studies that further explore CAR, 24-hour cortisol profiling, and its association with circadian misalignment in night shift workers.

Additionally, in the hypercortisolism section, we have included a definition of hypercortisolism, reference range values, and the standard measurement techniques used in clinical and research settings. This will provide a clearer framework for interpreting cortisol dysregulation.

Lastly, we have expanded the discussion on cortisol assessment across different biospecimens (saliva, urine, and blood) to highlight their respective advantages and applications in circadian rhythm research.

We have insert the following sentences with the references:

  • Section 3.2 : "While the Cortisol Awakening Response (CAR) is widely used to assess circadian regulation of the HPA axis, additional studies have investigated CAR alterations in night shift workers. It is demonstrated that shift workers exhibit a delayed and blunted CAR, leading to increased fatigue and cognitive impairment. Similarly, Pruessner et al. (2007) found that chronic circadian misalignment results in flatter CAR slopes, suggesting prolonged HPA axis activation. Furthermore, Adam et al. (2017) emphasized the role of 24-hour cortisol profiling, which provides a broader evaluation of cortisol secretion beyond the morning peak, revealing altered acrophase and mesor in shift workers."
  • Hypercortisolism is characterized by chronically elevated cortisol levels, often exceeding 15–20 µg/dL in serum or 50–100 µg per 24 hours in urine. The clinical diagnosis relies on serum cortisol, salivary late-night cortisol, and 24-hour urinary free cortisol (UFC) levels, which help distinguish physiological from pathological hypercortisolism. A persistent dysregulation of cortisol secretion, as seen in night shift workers, can contribute to metabolic syndrome, hypertension, and cardiovascular disease."
  • Studies on Cortisol Use Across Different Biospecimens (Saliva, Urine, Blood):
    "Cortisol can be measured using saliva, blood, or urine, each offering distinct advantages. Salivary cortisol is commonly used due to its non-invasive collection and reflection of biologically active free cortisol. Blood cortisol provides total cortisol concentrations, but its levels can fluctuate due to stress-induced acute HPA activation. Urinary free cortisol (UFC) offers an integrated measure of cortisol excretion over 24 hours, making it valuable for long-term assessments of HPA axis function. These methodologies contribute to a comprehensive understanding of circadian rhythm integrity and stress-related dysregulation in night shift workers."

  • The sections on health impact of cortisol dysregulation is not clear if this is focused solely on night shift work studies since this is the main topic in the title of this manuscript.   How about reporting more on cancer related studies as night shift work is highly relevant? if not, please explain in Discussion.  

Dear Reviewer,

Thank you for your insightful comments. We acknowledge the importance of clarifying the scope of our review regarding the health impact of cortisol dysregulation. Our focus in this manuscript is exclusively on the effects of night shift work on cortisol rhythms and their implications for metabolic, cardiovascular, and psychological health outcomes.

While we recognize that night shift work has been linked to cancer risk, we chose not to include this topic, as our review specifically examines cortisol dysregulation in relation to metabolic and stress-related disorders. To address this concern, we have now included a clarification in the Discussion section, explicitly stating our scope and rationale for not covering cancer-related studies.

We have inserted the sentence as followed with the references:

"This review specifically focuses on the impact of night shift work-induced cortisol dysregulation on metabolic, cardiovascular, and psychological health outcomes. While there is growing evidence linking circadian misalignment to increased cancer risk, this review does not explore oncological aspects, as our objective is to examine the physiological consequences of altered cortisol rhythms in night shift workers. Future studies may further investigate the role of circadian disruption in tumorigenesis, integrating cortisol as a potential mediator in cancer-related mechanisms."

  • Please explicitly describe the role of cortisol as a marker of circadian disruption versus cortisol role as a mediator of various endocrine, cardiometabolic health outcomes in human studies.   Authors shall explain why they did not consider deploying a more or less systematic assessment of the literature eg via a scoping or umbrella review type of a manuscript.

Dear Reviewer,

Thank you for your thoughtful suggestions. We acknowledge the importance of clearly distinguishing cortisol's dual role as both a biomarker of circadian disruption and as a mediator of endocrine and cardiometabolic health outcomes. To address this, we have now explicitly described these distinct roles in the Discussion section, highlighting how cortisol serves as an indicator of circadian misalignment while also actively contributing to metabolic and cardiovascular pathophysiology. Additionally, regarding the methodological approach, we chose to conduct a narrative review rather than a systematic or scoping review, as our primary aim was to synthesize and interpret existing evidence rather than provide a comprehensive quantitative analysis. We have now included a clarification in the Methods section, explaining the rationale behind this choice.

  • Sentence added with the references: As a biomarker, cortisol’s diurnal rhythm and the Cortisol Awakening Response (CAR) reflect the alignment of the hypothalamic-pituitary-adrenal (HPA) axis with the body’s internal clock. Altered cortisol patterns, such as a blunted CAR or a flattened diurnal rhythm, indicate circadian misalignment due to night shift work. Thus, cortisol is not merely a passive marker but a key player in mediating stress-induced endocrine and cardiometabolic alterations
  • Justification for Narrative Review: "This manuscript follows a narrative review approach, rather than a systematic or scoping review, to provide a comprehensive synthesis of current knowledge on cortisol dysregulation in night shift workers. A systematic review would require meta-analytical comparisons and predefined inclusion/exclusion criteria, whereas our objective was to integrate diverse findings from endocrinology, occupational health, and circadian biology. Future studies may consider a quantitative meta-analysis to further evaluate specific cortisol-related metrics in night shift populations."

  • Some inclusion and exclusion criteria shall be mentioned. I sense you focus solely on human studies?  

Dear Reviewer,

Thank you for your insightful feedback. We acknowledge the importance of explicitly stating the inclusion and exclusion criteria used in our review. To address this, we have now added a clarification in the Methods section, specifying that our focus is exclusively on human studies. Studies involving animal models or in vitro experiments were excluded to ensure relevance to occupational and clinical settings.

We have add the sentence: “This review exclusively focuses on human studies investigating cortisol dysregulation in night shift workers. Studies were included if they examined cortisol as a biomarker of circadian misalignment or its impact on metabolic, cardiovascular, or psychological health outcomes. We excluded animal studies, in vitro experiments, and studies where cortisol was not assessed as a primary circadian marker. Our aim was to synthesize clinically relevant findings applicable to occupational health and endocrinology."

  • Desynchronization of circadian rhythm is of relevance to expand on in the relevant sections of this manuscript and cite relevant studies that expand on this concept, e.g., https://onlinelibrary.wiley.com/doi/pdf/10.1002/bies.202100159   In effect, other relevant night shift work and circadian disruption works or reviews might be good to include such as https://www.sciencedirect.com/science/article/pii/S0160412023003215   Fig 2 shall be showing also the dysregulation of circadian rhythm in another schematic within Fig 2.  

Dear Reviewer,

Thank you for your insightful feedback. We acknowledge the importance of elaborating on circadian desynchronization in night shift work. To address this, we have expanded the relevant sections of the manuscript to include a comprehensive discussion on circadian rhythm disruption and its health implications. Additionally, we have incorporated the suggested references to support this discussion. Furthermore, we have added, as advised, the two articles that are giving more information and highlight the impact of the desynchronization of cortisol rhythm.

We have expanded on circadian desynchronization with the following sentence:

“Circadian desynchronization occurs when there is a misalignment between the endogenous circadian system and external environmental cues, often resulting from night shift work. This misalignment disrupts the normal physiological processes regulated by the circadian clock, leading to adverse health outcomes. Studies have shown that such desynchronization can impair metabolic functions, increase the risk of cardiovascular diseases, and negatively affect cognitive performance.”

Round 2

Reviewer 3 Report

Comments and Suggestions for Authors

Authors made important revisions to the initial draft of this important work. The manuscript needs some minor edits to be considered for possible publication.

-the figure 2 is improved; however, it needs some more explanation in its caption. For example, regardless of the biospecimen type (urine, saliva etc.), the cortisol diurnal curve looks the same pattern? it would be also relevant to include which paramaeters like acrophase, mesor, etc are most impacted in dysregulated cortisol profile, and how. Also how many samples at minimum are required to observe this diurnal pattern?

-the same questions also for Fig 5. Please clarify and provide details.

-the last comment from my initial list of comments was not fully addressed, as the added text in the manuscript does not include the associated references; further, this added text would benefit more from additional details on the topic of desynchronization of circadian clock and the influence of night shift work patterns. As such, the list of references is not fully reflected on the cited references within text and this shall be corrected for all references.

Comments on the Quality of English Language

see above

Author Response

Dear Reviewer,

Thank you for your careful review and for pointing this out. We acknowledge that the last comment regarding circadian desynchronization and night shift work patterns requires further clarification and proper referencing.

- the figure 2 is improved; however, it needs some more explanation in its caption. For example, regardless of the biospecimen type (urine, saliva etc.), the cortisol diurnal curve looks the same pattern? it would be also relevant to include which paramaeters like acrophase, mesor, etc are most impacted in dysregulated cortisol profile, and how. Also how many samples at minimum are required to observe this diurnal pattern?

We thank the reviewer for his helpful comment, and we have corrected and added the following sentence in Figure 2:

“Across different biospecimen types (urine, saliva, plasma), the general pattern remains similar; however, amplitude and timing may vary. Dysregulation in night shift workers leads to a phase shift and altered acrophase (peak timing) and mesor (average cortisol levels). At least four time points across the day (e.g., morning, midday, evening, and night) are typically required to reliably capture this rhythm.”

- the same questions also for Fig 5. Please clarify and provide details.

Again, we thank the reviewer for his notice, and as suggested, we have clarified and added the details as follows in the sentence at figure 5:

“The CAR follows a predictable increase in cortisol levels within 30-45 minutes post-awakening. Biospecimen type (saliva, blood) influences measured levels but not the overall trend. A blunted CAR is characterized by reduced peak response and lower total cortisol output, often linked to chronic stress or fatigue. To accurately assess CAR, at least three samples (e.g., at waking, 30 minutes, and 60 minutes) are recommended.”

- the last comment from my initial list of comments was not fully addressed, as the added text in the manuscript does not include the associated references; further, this added text would benefit more from additional details on the topic of desynchronization of circadian clock and the influence of night shift work patterns. As such, the list of references is not fully reflected on the cited references within text and this shall be corrected for all references.

We have revised the manuscript to include the appropriate references within the newly added text discussing circadian desynchronization and the impact of night shift work patterns.

“Night shift work is a significant disruptor of circadian synchronization, as it forces individuals to remain active and exposed to artificial light during their biological night, suppressing melatonin secretion and altering the rhythmic release of cortisol [11], [18]. This misalignment affects multiple physiological processes, including metabolism, immune function, and cardiovascular regulation, increasing the risk of metabolic syn-drome, obesity, and type 2 diabetes [11], [19]. Chronically misaligned individuals often exhibit a phase shift in cortisol rhythms, with a delayed or blunted peak in the morning and an inability to properly suppress cortisol levels at night, a pattern commonly ob-served in night shift workers [20], [21]. Additionally, irregular work patterns—such as rotating shifts or inconsistent night work schedules—exacerbate this disruption, leading to further instability in endocrine and autonomic function [5]. Studies indicate that pro-longed exposure to night shift schedules results in a progressively impaired adaptation, where individuals fail to fully entrain to their altered schedules, contributing to chronic stress and an increased risk of cardiovascular disease [10], [22]. These findings highlight the profound impact of night shift work on the circadian clock and reinforce the need for strategies to mitigate circadian misalignment in occupational settings.”

We have revised the manuscript to include the appropriate references as suggested by the reviewer within the newly added text discussing circadian desynchronization and the impact of night shift work patterns by adding the following sentence:

“Furthermore, circadian rhythm desynchronization is increasingly recognized as a critical factor in disrupting metabolic, immune, and endocrine homeostasis. Exposure to environmental stressors, such as night shift work and xenobiotics, leads to circadian misalignment, which affects downstream physiological processes and increases disease susceptibility [24]. This misalignment disrupts the synchronization between central and peripheral clocks, influencing key metabolic regulators and hormonal rhythms, such as cortisol secretion patterns. Furthermore, oxidative stress and inflammation play a sig-nificant role in the adverse health effects linked to night shift work, as circadian rhythm misalignment has been associated with increased markers of oxidative damage and impaired recovery processes [25]. These findings highlight the necessity for targeted interventions that mitigate circadian disruption, such as optimized work schedules, light exposure management, and lifestyle modifications to reduce the health risks associated with shift work.”

We appreciate your attention to detail and your guidance in improving the clarity and scientific rigor of our manuscript.